# Oxidative Stress of Cadmium and Lead at Environmentally Relevant Concentrations on Hepatopancreas of *Macrobrachium nipponensis* and Their Mixture Interactivity: Implications for Water Quality Criteria Amendment

**DOI:** 10.3390/ijerph20010360

**Published:** 2022-12-26

**Authors:** Xiang Liu, Qianzhen Deng, Hao Yang, Jingyao Wang, Min Wang

**Affiliations:** 1College of Agricultural Science and Engineering, Hohai University, Nanjing 210098, China; 2Anhui Provincial Key Laboratory of Environmental Pollution Control and Resource Reuse, Anhui Jianzhu University, Hefei 230009, China; 3Beijing Municipal Research Institute of Environmental Protection, Beijing 100037, China; 4National and Local Joint Engineering Research Center of Ecological Treatment Technology for Urban Water Pollution, Wenzhou University, Wenzhou 325035, China

**Keywords:** cadmium, lead, biomarker, interaction effect, hepatopancreas, *Macrobrachium nipponensis*

## Abstract

The biotoxicity of heavy metals in water has always been the focus of ecological health research. In this study, the oxidative stress-associated toxicity of cadmium (Cd) and lead (Pb) at environmentally relevant concentrations on the hepatopancreas of *Macrobrachium nipponensis* was investigated based on multiple biomarker responses in a 28-day indoor exposure study. Changes in integrated biomarker responses (IBR) and their interactivity were subsequently analyzed. No dead individuals were found across any of the tested conditions. The chronic toxicity of heavy metals depended on their type and exposure time at the same concentration. At low concentrations, organisms have a regulatory capacity to cope with the excess reactive oxygen species (ROS) induced by Pb stress over time. In detail, the activity of superoxide dismutase (SOD) was inhibited by Pb stress at a high concentration as time passed. The sensitivity of metallothionein (MT) to Cd was stronger than Pb, and the potential for Cd to cause lipid peroxidation damage was higher than Pb. At the same time, Pb had a greater disturbance effect on the nervous system than Cd, especially in the early exposure stage. The contribution of Cd and Pb to the interaction effect varied dynamically with time and concentration of exposure, but mostly showed antagonism. The results of this study have important significance for guiding the diagnosis of ecological water health, the amendment of water quality criteria, and the management of wastewater discharge.

## 1. Introduction

In recent decades, a large amount of toxicant-rich wastewater and surface runoff has been randomly discharged into water bodies due to ongoing urbanization and industrialization. Moreover, the biogeochemical cycles of contaminants are altered because of hydrodynamic changes arising from the overexploitation of water resources [1]. Regional pollution has become a burning issue in regulated rivers. Heavy metals are considered one of the most hazardous and bioaccumulative contaminants in aquatic environments [2,3]. In view of their lipophicity, persistence and toxicity characteristics, heavy metals may be easily enriched in organisms and further scaled up in aquatic food chains. Consequently, heavy metals ultimately pose potential threats to the stability of water ecosystems and even to human health [4]. Therefore, a comprehensive understanding of metal-induced chronic toxicity at environmentally relevant concentrations to living organisms is of great significance in amending water quality criteria.

The Huai River basin (HRB), one of the main grain-producing areas of China, has been severely polluted by the discharge of agricultural non-point sources, livestock breeding and industrial wastewater [5,6]. In this regard, cadmium (Cd) and lead (Pb), two typical heavy metals with substantial toxicity and widespread distribution, have unexceptionally entered the HRB water column, presenting considerable concentrations [7]. As early as 1976, they were listed by the Environmental Protection Agency of the United States as priority pollutants [8]. The primary biological effects of Cd are to stimulate the respiratory tract and harm the liver or kidneys of exposed organisms after accumulation [9]. Pb has a toxic impact on the nervous system and can be easily accumulated in the blood or bone to produce pernicious effects [10]. Additionally, both substances may be absorbed by organisms and combined with proteins, enzymes, and other macromolecules through anion transport channels to produce irreversible denaturation. As a result, changes in physiological and metabolic processes or genetic information mutations occur [11]. More importantly, these substances may be chelated with organics in the water to form metal-organic compounds, exhibiting more profound environmental risks [12].

Sediments may be unavoidably transferred by heavy metals discharged into water bodies by adhering to the surface of suspended particles. They can also be released into overlying water under hydrodynamics and bioturbation. At the sediment–water interface, the concentrations of metal ions can be maintained at dynamic equilibrium, resulting in multiple risks to benthic animals living in this active region. Benthic macroinvertebrates are a group of animals with many different classes that connect the preceding and the following ecological niche relating to material circulation and energy flow in the aquatic food chain [13,14]. Consequently, they are essential to the water ecosystem structure and its functional integrity. The oriental river prawn *Macrobrachium nipponensis* (*M. nipponense*), commonly known as freshwater prawn, is a typical benthic dominant Crustacea species ubiquitously found in rivers, lakes, and reservoirs throughout east Asian countries, particularly in China [15]. *M. nipponense* (Crustacea: Decapoda) can absorb the attached algae, organic particles, and litter as nutrients and can also be consumed by people or as high-quality food for high-trophic level organisms in the aquatic ecosystem. However, when heavy metals exceed a certain threshold in vivo, a large amount of reactive oxygen species (ROS) is generated. Therefore, a series of toxic reactions occur in *M. nipponense* individuals due to heavy metal contamination, such as abnormal behavior, physiological disorders, histopathological abnormalities, and even acute death in some extreme cases [16,17]. Heavy metal ions dissolved in water are continuously absorbed through the gills and transported to various body parts through blood or accumulated in surface cells. Second, during feeding, heavy metals in water or residues in bait enter the body through the digestive tract. Additionally, osmotic exchange between the surface of the body and water may also be essential for heavy metals entering the body. Hepatopancreas, a metabolic detoxification organ for foreign substances, has the highest degree of contact with toxic matters and is a critical organ for antioxidant defense [18].

Traditional chemistry method-based monitoring can achieve accurate quantification of Cd and Pb. However, it does not accurately reflect the bioavailability and synergistic or antagonistic toxic effects associated with other chemical contaminants. With the development of molecular biology-based technologies, researchers are gradually favoring biomonitoring, which is becoming an emerging technology in environmental monitoring [19]. The working principle of molecular biology-based technologies is to monitor abnormal changes on the molecular level of individuals and cells in the event of pollution stresses and, therefore, to characterize the pollution and toxic effects. Compared with traditional monitoring, molecular biology-based technologies are more scientifically accurate methods with the advantages of continuity and comprehensiveness. Once the concentration of essential and non-essential heavy metals exceeds a certain threshold in aquatic organisms, a large amount of ROS is generated, causing oxidative damage. Still, organisms have an antioxidant defense system to protect tissues and cells from oxidative damage and maintain the dynamic balance between the production of reactive oxygen species and redox in vivo [20]. The toxic effect of heavy metals interferes with normal physiological metabolism by activating or inhibiting enzymes or non-enzymatic substances in the antioxidant defense process. Previous studies have focused on the form distribution, transference, and ecological risk of heavy metals in sediments. However, minimal attention has been paid to the chronic toxicity of low-concentration heavy metals or multiple heavy metals combined. 

Acute and chronic Cd and Pb toxicities on aquatic organisms have been well documented at distinct levels. Still, to the best of our knowledge, few studies have systematically evaluated oxidative stress-related damage and the interactivity of these two metals in crustaceans at environmentally relevant concentrations. Therefore, this study aimed to investigate the oxidative stress effects of dissolved metal ions (Cd^2+^ and Pb^2+^), individually and as mixtures, on *M. nipponensis* exposed in the laboratory for 28 days. Multiple biomarkers were employed to comprehensively evaluate their chronic biotoxicity at various concentrations. The results presented here may have significant practical implications for heavy metal risk warning, water quality benchmarking, and watershed environment management.

## 2. Materials and Methods

### 2.1. Chemical Reagents and Solution Preparation

Cadmium acetate dihydrate (C_4_H_6_CdO_4_·2H_2_O) and lead nitrate (Pb (NO_3_)_2_) used in this study were of analytical grades. These substances were purchased from Sinopharm Chemical Reagent Co., Ltd., China. The corresponding stock solutions (100 mg/L) were prepared by dissolving the chemicals with deionized water according to approved procedures [17]. All glassware used in this step was completely immersed in nitric acid (10% *v*/*v* HNO_3_ AR grade, Merck, Rahway, NJ, USA) for 24 h and then rinsed thoroughly with ultrapure water (Milli-Q^®®^, 18 MΩ/cm, Merck).

### 2.2. Testing Organisms

The freshwater river prawn, *M. nipponense*, which was selected as the testing organism, was obtained from the Nanjing Aquatic Science Research Institute in China. Currently, prawns have been frequently used in laboratory-based ecotoxicological studies due to their high sensitivity to pollutants and the feasibility of indoor culture [17]. In this study, the selected individuals were initially acclimated to 40 L plastic buckets containing dechlorinated tap water for approximately one week. In this process, organisms were maintained at a temperature of 10 ± 1 ℃, pH 7.30 ± 0.08, ammonia nitrogen and nitrate <0.1 mg/L, and 12 h light: 12 h dark photoperiod. Additionally, the water was continuously aerated to maintain a dissolved oxygen concentration of 6.20 ± 0.40 mg/L. After acclimatization, 140 healthy prawns with homogeneous body sizes (average weight: 2.62 ± 0.15 g, average length: 6.43 ± 0.21 cm) were randomly assigned to seven groups, including 20 individuals each, for subsequent exposure tests.

### 2.3. Experimental Design

First, 25 L of dechlorinated tap water was added to each bucket. Subsequently, a certain amount of stock solution was added to the bucket based on the concentration set earlier and mixed well. Afterward, the pH value of the water was determined and adjusted to 7.2–8.0. In this study, the whole experiment was designed into seven groups in total, including a control group and six experimental groups (Figure 1). The experimental groups contained single Cd, Pb, and their combined group, and each treatment group covered both low (0.01 mg/L) and high (0.10 mg/L) concentrations. The exposure protocol was as follows: control (0.00 mg/L Cd^2+^, 0.00 mg/L Pb^2+^), low group (0.01 mg/L Cd^2+^, 0.01 mg/L Pb^2+^, and 0.01 + 0.01 mg/L Cd^2+^+ Pb^2+^) and high group (0.10 mg/L Cd^2+^, 0.10 mg/L Pb^2+^, and 0.10 + 0.10 mg/L Cd^2+^+ Pb^2+^). Eventually, each group of 20 prawns was delivered to each experimental bucket with a certain amount of *Elodea* in advance. A chronic toxicity test for 28-day exposure was conducted under the designed conditions (10.0 ± 1.0 ℃, pH 7.20 ± 0.10 and a natural light/dark regime). The prawns were fed wheat bran with a commercial shrimp diet, once every three days. Mortality was checked daily, and prawns were considered dead when their shells gaped and failed to shut again after external stimuli.

### 2.4. Sampling and Biomarker Analysis

After the periods of 7, 14, 21 and 28 days, three *M. nipponense* individuals in each bucket were collected and dissected by cutting the cephalothorax with a tweezer and scalpel. The hepatopancreas of *M. nipponense* individuals was removed from each body. After that, the samples were washed with physiological saline solution (0.86% NaCl), blotted with filter paper, and stored at −80 °C. The samples were individually frozen and mechanically pulverized in a mill with liquid nitrogen for biomarker analysis. A suite of biomarkers of each homogenized sample including superoxide dismutase (SOD, U/mg protein), catalase (CAT, U/mg protein), glutathione Peroxidase (GPx, U/mg protein), metallothionein (MT, ng/mg protein), malondialdehyde (MDA, nmol/mg protein) and acetylcholinesterase (AChE, U/g protein) were determined using corresponding protocols described by the bioassay kits from Nanjing Jiancheng Bioengineering Institute, China.

### 2.5. Statistical Analysis

Biomarkers were expressed as mean ± standard deviation (SD), normality, and homoscedasticity tests for the obtained results using the Shapiro–Wilk and Levene methods in SPSS Statistics 22.0 (IBM Co., Chicago, USA). One-way analyses of variance (ANOVA) were used for statistical analysis between the groups. The differences between the experimental and control groups were analyzed using the Dunnett T3 test, and the significance level was set as *p* < 0.05.

Integrated biomarker responses (IBR) [21,22] was calculated based on the biomarkers tested in this study. The specific calculation formulas are as follows [23]. The IBR values of the different treatment groups were plotted as a star chart. A higher IBR value represents greater oxidative stress-related toxicity in the organism [24].
$$Y_{i}=(X_{i}-\bar{x})/S\;B_{i}{=Z}_{i}+\left|Y_{\min}\right|\;A_{i}=\frac{1}{2}\sin\frac{2\pi }{n}{\times B}_{i}{\times B}_{i+1}\;\mathrm{IBR}=\sum_{i=1}^{n}\frac{A_{i}}{n}$$
where $X_{i}$ and $\bar{x}$ represent the mean value of a biomarker from each treatment group and all the groups, respectively; *S* represents the standard deviation of each biomarker; $Z_{i}$ was calculated as +$Y_{i}$ when activated and −$Y_{i}$ when inhibited; $\left|Y_{\min}\right|$ represents the absolute value of the minimum value after standardized treatment for each biomarker; $B_{i}$ and $B_{i+1}$ represent the individual biomarker scores, and *n* represents the number of biomarkers.

SPSS statistical software (ver. 22.0, SPSS Company, Chicago, IL, USA) was used to analyze the interaction of Cd-Pb at different concentrations on organisms by 2 × 2 factorial analysis of variance. The interactivity of Cd and Pb on the organisms could be determined according to the dose-effect curve obtained from the above analysis. There two dose–effect curves would parallel each other if there were no interactions between these two metals, showing an additive effect. The corresponding interaction effects would be synergistic and antagonistic if these two dose–effect curves gradually moved away or intersected as the concentration of pollutants increased, respectively.

## 3. Results

### 3.1. Mortality

No dead individuals were found under all tested conditions after 28 days of exposure.

### 3.2. The Responses of Antioxidant Biomarkers to the Tested Conditions

Changes in SOD, CAT and GPx activities in the hepatopancreas of *M. nipponense* under single and combined Cd and Pb stresses were shown in Figure 2. As can be seen in Figure 2A, the initial inhibitory effect of single Cd on SOD activity was enhanced as the concentration increased. SOD activity gradually recovered to a level with no significant difference from the control group as the exposure time extended. However, the recovery of SOD activity was significantly inhibited by the relatively high concentration of Pb (*p* < 0.05) compared to the low concentration group. The changing trend of SOD activity with exposure time was consistent with that of Pb alone when Cd and Pb were co-exposed. Remarkably, after 21 days of culture, SOD activity was significantly activated (*p* < 0.05). In all test groups, CAT activity was activated by heavy metal stress, especially reaching the maximum value on the 14th day of exposure.

Meanwhile, at 14 days, the effect of single Pb on CAT activity in the hepatopancreas was positively correlated with increased concentration, while that of single Cd was inversely correlated. This effect was changed by combined exposure, which had a significant positive effect. Concerning GPx activity, the effect of single Cd on it in the hepatopancreas was completely different from that of single Pb. Specifically, under the stress of a single Cd, the activity of GPx was significantly inhibited in the initial stage compared with the control group. Still, on the 21st day of exposure, the activity of GPx was restored to the level of the control group. On the 28th day, the activity of GPx was significantly inhibited again, becoming less than half that of the control group. Under single Pb stress, activity was inhibited at each point. The inhibition was most significant on the 7th and 21st days of exposure, and the whole change pattern was positively correlated with the concentration change. The inhibition of Pb on GPx activity was weakened by combined exposure.

### 3.3. The Responses of Specific Biomarkers to the Tested Conditions

The changes of specific biomarkers, including MT, MDA, and AChE, in the hepatopancreas of *M. nipponense* were shown in Figure 3 after the 28th day of individual and combined exposure to Cd and Pb. Compared with the control group, MT levels in cells were enhanced to varying degrees by Cd and Pb, either alone or in combination (Figure 3A). Particularly, MT contents were significantly increased under single Cd exposure and had a positive concentration effect, reaching a maximum of 0.88 ± 0.08 ng/mg protein after 28 days at a concentration of 0.1 mg/L. At the same time, from the time scale, relatively high values appeared on the 14th and 28th days of exposure, which were significantly higher than the ones at other time points (ANOVA, *p* < 0.05). However, for single Pb exposure, MT contents had no significant differences in the time and concentration scales, although Pb exposure still had a significant increase in trends compared to the control group (ANOVA, *p* < 0.05). The changing trends of MT contents over time and concentration scales were significantly altered when Cd and Pb were co-exposed at the same concentrations. The relatively high values were mainly in the middle stage of the entire exposure, especially at high concentrations.

According to the MDA content in the hepatopancreas, Cd, and Pb alone or combined could cause different degrees of oxidative damage in the cells (Figure 3B). Specifically, under single Cd stress, MDA content in the hepatopancreas had positive effects on time and concentration, reaching 22.16 ± 0.95 nmol/ng protein after 28 days at 0.10 mg/L. However, the MDA content gradually decreased with prolonged exposure time under single Pb stress. This result was the opposite of that of single Cd exposure. MDA content increased at a low concentration (0.01 mg/L) as time passed when Cd and Pb were exposed together. The maximum value was reached at a concentration of 0.10 mg/L after 14 days. This result was about 3.0 times that of the control group.

As shown in Figure 3C, Cd and Pb, either alone or combined, had both activating and inhibiting effects on the activity of AChE in the hepatopancreas of M. nipponense. For example, AChE was significantly impacted by Cd concentration when exposed to Cd alone. Compared with the control group, its activity was significantly inhibited at 0.01 mg/L and 0.10 mg/ L after 14 and 21 days of exposure, respectively. Still, it was significantly activated at the end of exposure (ANOVA, *p* < 0.05). The activation effect occurred only in the early exposure stage when cells were exposed to Pb alone, regardless of the concentration difference. However, the activity of AChE was gradually restored with the increase of Pb concentration as time passed. It was significantly activated in the second half of the exposure period (ANOVA, *p* < 0.05). The activation effect of Pb on AChE at the initial exposure stage was significantly attenuated by co-exposure to Cd and Pb at the same concentrations. Still, there was no change in the activation effect of Pb at high concentrations.

### 3.4. Chronic Biotoxicity Assessment of Metals Tested and Their Interaction Patterns Analysis

As seen in Figure 4, chronic biotoxicity to varying degrees was caused by Cd and Pb at environmentally relevant concentrations, either alone or combined, to hepatopancreas of *M. nipponense*. Specifically, the oxidative toxicity caused by single Cd stress had a remarkable time- and dose-dependent effects on the cells. The IBR value varied from 0.38 to 2.36, with the minimum occurring at 0.01 mg/L Cd after seven days and the maximum occurring at 0.10 mg/L Cd after 28 days. In contrast, the chronic biotoxicity induced by single Pb stress still had a concentration effect, but it was manifested in the early exposure stage. Furthermore, except for the co-exposure group with high concentrations, the oxidative toxicity of the other groups was low after 14 days of exposure. Additionally, the oxidative toxicity caused by the combined metals was extremely significant under high concentration stress, covering a more extended exposure period. The IBR value under this condition reached a maximum of 2.51 after 21 days of exposure and decreased to 1.21 after 28 days of exposure.

The interaction analysis of Cd and Pb at different concentrations on the hepatopancreas of *M. nipponense* is shown in Table 1, and the interaction types are shown in Figure 5. The efficacy of a single metal on the IBR values varied with exposure time and concentration in the combined exposure group. According to the Partial Eta^2^ values of each group (Table 1), the contribution of Pb to IBR was higher than that of Cd at both low and high concentrations after seven days of exposure. After 14 days of exposure, Cd was more effective than Pb at low concentrations, while Pb was more effective than Cd at high concentrations. Over time, the contribution of Cd and Pb to the final IBR values was equal to 21 days after exposure to higher Cd than Pb at the end of the exposure. However, the interaction pattern of these two metals did not change significantly with exposure time and concentration. The overall interaction had an antagonistic effect and showed only a weakly additive impact 14 days after exposure to high concentrations (Figure 5).

## 4. Discussion

### 4.1. Effects of Cd and Pb at Environmentally Relevant Concentrations on Biomarkers Associated with Oxidative Stress in the Hepatopancreas of M. nipponense

As a general rule, the hepatopancreas of river prawns is often regarded as a detoxification organ, as it contains a variety of ion transferases, detoxification enzymes and protective enzymes [25]. It is also the most active organ for heavy metal accumulation and oxidative stress responses. The data showed that the concentration of Cd and Pb enriched in the hepatopancreas was proportional to the concentration in water [17]. Oxidative stress may be induced by metal ions enriched in the hepatopancreas can induce oxidative stress by activating xanthine oxidase and heme oxidases, or blocking the respiratory chain, electron transport chain and enzymatic reactions, ultimately leading to injury or even the death of individuals [26]. Among them, Cd induces oxidative stress by increasing lipid peroxidation or altering intracellular glutathione levels, while Pb interferes with membrane-related processes through transverse phase separation, including the activity of membrane enzymes, solute trans-bilayer transport and signal transduction processes [27].

SOD is the first redox metalloenzyme to interact with ROS. It is distributed in the hepatopancreas and can rapidly decompose O_2_- into H_2_O_2_ and O_2_ [28,29]. The hepatopancreas of *M. nipponense* could eventually restore SOD activity to the level of the control group under stress conditions caused by Cd alone. This indicated that the in vivo antioxidant defense system had a specific ability to regulate the generated ROS, which might also be caused by a metabolic compensation mechanism [30]. However, the SOD activity of hepatopancreas was consistently inhibited under high Pb concentration stress. This was because Zn^2+^ or Mn^2+^ tends to be replaced by Pb ions in Cu/Zn-SOD in Mn-SOD [31], respectively. This leads to a change in the SOD structure and decreased activity. On the other hand, heavy metal ions could combine with the -SH group in the enzyme molecule, reducing SOD activity [32]. Catalyzation of the decomposition of H_2_O_2_ into H_2_O and O_2_ would continue to be made by CAT to prevent excessive accumulation of H_2_O_2_ from causing damage to the organism [33]. It could be seen from the results that CAT activity in the hepatopancreas was activated under Cd and Pb stresses. This result confirmed that the organism produced excess H_2_O_2_. However, its activity did not increase linearly on the time scale, and the main reason might be related to the enrichment course of metal ions in vivo. Second, CAT activity is also significantly regulated by SOD activity, presenting compensatory feedback for inhibiting SOD activity. Additionally, H_2_O_2_ might be reduced to H_2_O by GPx by using glutathione (GSH) as an electron donor [34] and simultaneously generating l-glutathione oxidized (GSSG). GSSG may be reconverted into GSH via the catalysis of glutathione reductase (GR) [28]. Changes in GSH content in organisms are crucial in regulating GPx activity. In this study, the overall GPx activity in the hepatopancreas was inhibited by Cd and Pb ions to varying degrees. This was mainly due to the oxidative stress effect caused by heavy metal ions disrupting the intracellular redox balance conditions and changing the GSH: GSSG ratio, resulting in insufficient electron donors for GPx to convert H_2_O_2_ [35].

ROS may be removed by MT by switching between a hydroxyl reduction state and an oxidation state [36]. The bioavailability of heavy metal ions in cells may be reduced to a certain extent by MT, which plays a significant detoxification role. Here the MT content in the hepatopancreas of *M. nipponense* was significantly higher than that of the control group under all experimental conditions. However, such increase patterns were different. The main reason for this was that the accumulation of heavy metal ions in the organism exceeded the binding capacity of MT, with an increase in heavy metal concentration as time passed [37]. Consequently, excess heavy metal ions exist in a free state, having significant toxic effects on the organism and severely inhibiting the synthesis of metal-binding proteins, including MT. It has been reported that heavy metal ions can induce the expression of MT within a specific concentration range, and the amount of MT synthesis is proportional to the concentration of heavy metals [38]. However, when the concentration is above the limit, metal ions have toxic effects on organisms as they may replace metals or functional groups in enzyme molecules, leading to the inactivation of the enzyme. This may be why the MT content in the hepatopancreas only had a linear response under a particular exposure concentration and time range. 

Lipid peroxides (LPO) is considered to be a significant mechanism of heavy metal oxidative toxicity [39]. As one of the critical products of the LPO reaction, MDA can effectively indicate changes in oxidative damage, cell dysfunction, and physicochemical properties of cell membranes. The MDA content in the hepatopancreas of the freshwater shrimp under Cd and Pb stress in the water body was significantly higher than that of the control group, and its content changed dynamically with exposure time. However, there was no apparent time- and dose-dependent effect for the treatments with Pb stress alone. It was indicated by these results that there were differences in the action of modes of different metal ions on LPO. The reason for this phenomenon was that ROS generated by heavy metal stress could not be removed over time, and the double bonds of unsaturated fatty acids in membrane phospholipids were attacked by excess ROS. Consequently, this resulted in lipid peroxidation, and the MDA content increased accordingly [40].

Meanwhile, metal ions can combine with negatively charged groups on the membrane surface, affecting the charge density and local electrostatic field on the membrane surface, changing the dielectric constant, and causing LPO damage [41]. Second, the generated MDA can also be cross-linked with NH_2_ on the membrane to form Schiff’s bases, aggravating oxidative damage [42]. The heavy metals Cd and Pb also had varying degrees of inducing and inhibitory effects on AChE activity in the hepatopancreas at different exposure concentrations and times from a neurotoxicity perspective. For example, after exposure to the low, medium, and high concentrations of Pb for seven days, its activity was significantly higher than that of the control group. After 28 days of exposure at low concentrations, its activity was significantly inhibited. On the other hand, the enzyme molecule contains a large amount of -SH, which can combine with heavy metals to cause spatial conformational changes and indirectly reduce enzyme activity [43]. However, the inducing effect of AChE activity in specific periods might be due to the binding of heavy metals and acetylcholine receptors under the stimulation of pollutants. This interaction reduces the binding efficiency of AChE to its reception [44]. Organisms may promote AChE synthesis to degrade neurotransmitters, as a certain degree of adaptation is shown by regulatory mechanisms.

### 4.2. Combined Toxicity of Cd-Pb and Its Interaction Mechanisms

As a visual evaluation tool for chronic biological toxicity, IBR can effectively avoid the inaccuracy of evaluation results due to the different responses of individual biomarkers to pollutants. According to the IBR values, the differences in the biological chronic oxidative toxicity of different heavy metals in water can be identified at different concentrations and times. It was revealed in a previous study that *Limnodrilus hoffmeisteri* showed some adaptive ability to perfluorooctane sulfonic acid (PFOS), Zn alone, and their combination according to IBR values [45]. Furthermore, the changes in SOD, CAT, and LPO contents in *Mytilus galloprovincialis* in the Bay of Saronikos in Greece successfully described the heavy metal pollution status and ecological risks of coastal waters using the IBR values [46]. In this study, the IBR values of each treatment group had an irregular trend with the prolongation of exposure time. Still, after 10 days of exposure, the IBR values of all groups decreased, also showing a specific adaptive ability, which followed previous reports. Six molecular biomarkers in the hepatopancreas of *Macrobrachium japonicus* were comprehensively calculated, and the “causal effect” between different heavy metal treatment groups and organisms was established. Both individual and combined exposures to Cd and Pb at different concentrations and after different treatment times resulted in various degrees of oxidative toxicity to the hepatopancreas. These were consistent with previous results when dynamic changes were observed. For example, the chronic biotoxicity of Cd-only treatments gradually increased over time, while the chronic biotoxicity of Pb-only treatments and the combination of the two decreased with time. This indicates that the hepatopancreas of *M. nipponense* can conduct a detoxification function on a time scale through its antioxidant regulation during heavy mental stress.

The combined and simultaneous toxicity of multiple heavy metals acting on an organism was far more complicated than that of a single heavy metal. The species, toxic ratio, exposure time, and metal form are all key factors affecting the interaction. The uptake of heavy metals by organisms and their ability to compete with binding sites determine the type of interaction between heavy metals. Currently, there are three main types of interaction: additive, synergistic, and antagonistic. The combined interaction between factors changes with increased heavy metal exposure concentration and the prolongation of time. It has been shown in recent studies that the primary mechanism of Cd-Zn antagonism was that Zn inhibits the absorption of Cd or accelerates the transport of ingested Cd to internal organs. In addition, as they have a similar structure, they compete with transporter-binding sites. In this study, the interactive effect of Cd and Pb combined exposure on the hepatopancreas of *M. japonicus* had identical changes to those reported in previous studies, displaying an overall antagonistic impact. This antagonism may be explained by the site competing theory [47]. Before entering the cell, metal ions must first bind to the active site on the cell surface. When heavy metal ions coexist, site competition occurs, and each ion interferes with the other for such binding. Therefore, overall biological toxicity is reduced.

Altogether, the mechanism of the combined toxicity of heavy metals in the water is complex for aquatic organisms. It cannot be inferred that the combined toxicity of heavy metals to organisms must be greater than that of a single heavy metal, nor can it be straightforwardly deduced that the combined toxicity is the addition of the toxicity of every single metal. The combined toxicity of any two or other heavy metals must be determined by specific exposure experiments. Different heavy metals, different organisms, different concentrations, and exposure times are all essential factors contributing to toxicity and interactive effects.

### 4.3. Implications for Water Quality Criteria Amendment and Environmental Remediation

Water quality security is related to ecological health, human survival, and development. Establishing complete and accurate water quality criteria is a basis and an essential condition to effectively manage the water environment in a river basin. However, each pollutant discharged into water often has various biological effects, which often have complex interaction effects with other coexisting pollutants. Therefore, the traditional method of evaluating water quality safety with the help of pollutant concentrations has specific inaccuracy. Biological monitoring can directly reflect the comprehensive toxicity of target pollutants in water, which has gradually developed into an essential supplement for water quality safety assessment and effectively makes up for the shortcomings of traditional physical and chemical testing.

It was made evident in this study that when environmental concentrations of Cd and Pb coexist, they present antagonistic interactions. This result has good enlightenment for water quality standards, especially for complex water environment systems, and water quality standards should not be considered only from the pollutants themselves. Additionally, aquatic organisms have an antioxidant defense system that has a specific detoxification function, and the toxicity of contaminants to aquatic organisms at environmental concentrations has a time effect. Therefore, the time effect of the toxic effect of pollutants should be considered comprehensively in revising water quality standards and remediation of environmental pollutants.

## 5. Conclusions

In conclusion, the oxidative stress-associated toxicity of Cd and Pb at environmentally relevant concentrations in the hepatopancreas of *M. nipponense* was dynamic on a time scale. Under equivalent concentrations, the chronic toxicity of single Cd stress was embodied at the end of the exposure. In contrast, for a single Pb, biotoxicity had a time-delay property. However, the toxicity showed positive concentration effects. When these two heavy metals coexisted, their interaction was antagonistic, regardless of the dose changes and exposure duration. The contribution of Cd and Pb to their interactions varied with the dose and duration of exposure. At 21 days of exposure, Cd and Pb contributed equally to their interaction effects, regardless of the concentration. Therefore, in the processes of water quality criteria amendment and water environment ecological restoration, the interaction effects of combined pollutants and their dynamic changes in time and concentration should be considered comprehensively to improve the scientific nature of basin water environment management.

## Figures and Tables

**Figure 1 ijerph-20-00360-f001:**
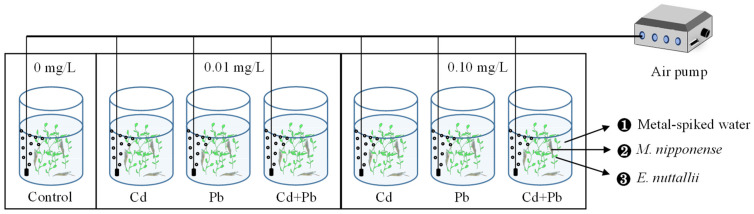
Schematic diagram of chronic toxicity tests (28 days) with *M. nipponense* exposed to two concentration gradients for water spiked with Cd, Pb and their mixture.

**Figure 2 ijerph-20-00360-f002:**
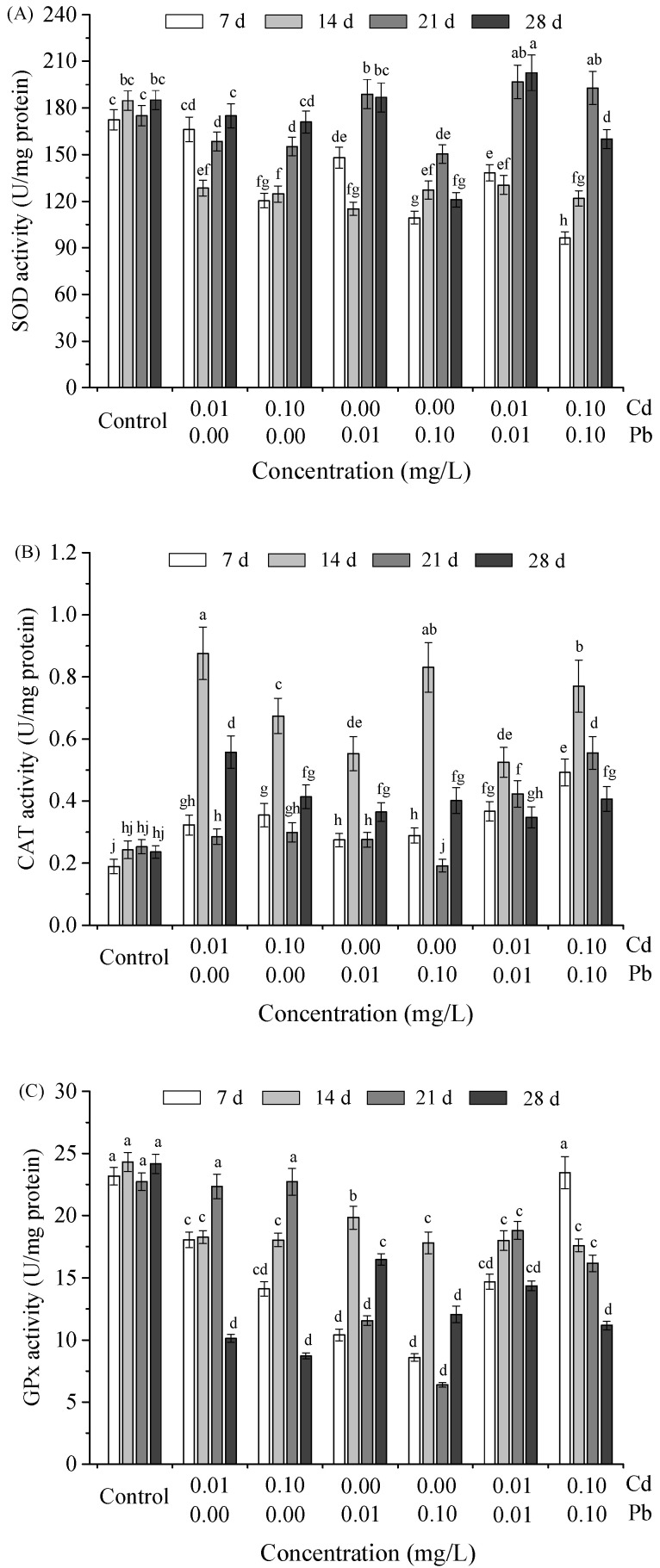
Changes of antioxidant enzyme activities including SOD (**A**); CAT (**B**); and GPx (**C**) in the hepatopancreas of *M. nipponense* following various treatments with single Cd, Pb and their mixture along increasing concentrations (0.01 and 0.10 mg/L) at four time points (7, 14, 21 and 28 days). Different letters indicate a significant difference among groups (ANOVA, *p* < 0.05), while the same letters indicated no significant difference (ANOVA, *p* < 0.05). Control represents the water without Cd and Pb.

**Figure 3 ijerph-20-00360-f003:**
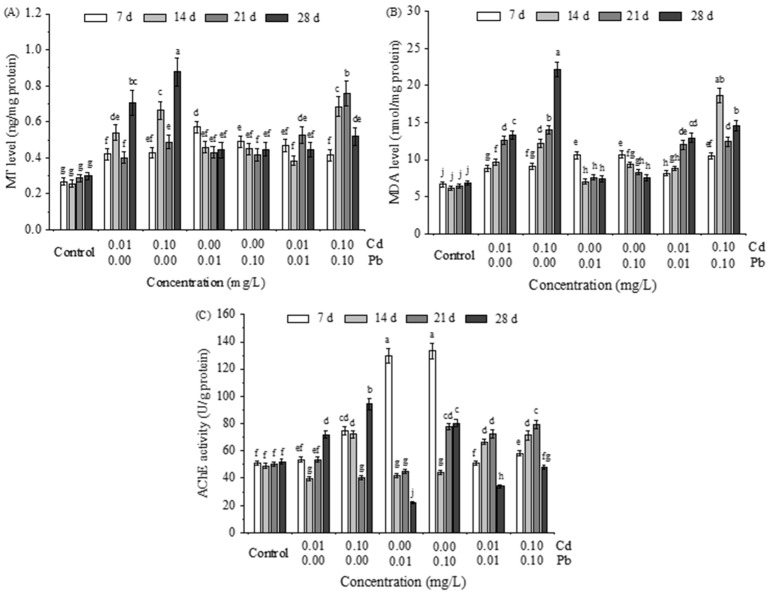
Changes of MT (**A**); MDA (**B**); and AChE (**C**) in the hepato of *M. nipponense* following various treatments with single Cd, Pb and their mixture along increasing concentrations (0.01 and 0.10 mg/L) at four time points (7, 14, 21 and 28 days). Different letters indicate a significant difference among groups (ANOVA, *p* < 0.05), while the same letters indicated no significant difference (ANOVA, *p* < 0.05). Control represents the water without Cd and Pb.

**Figure 4 ijerph-20-00360-f004:**
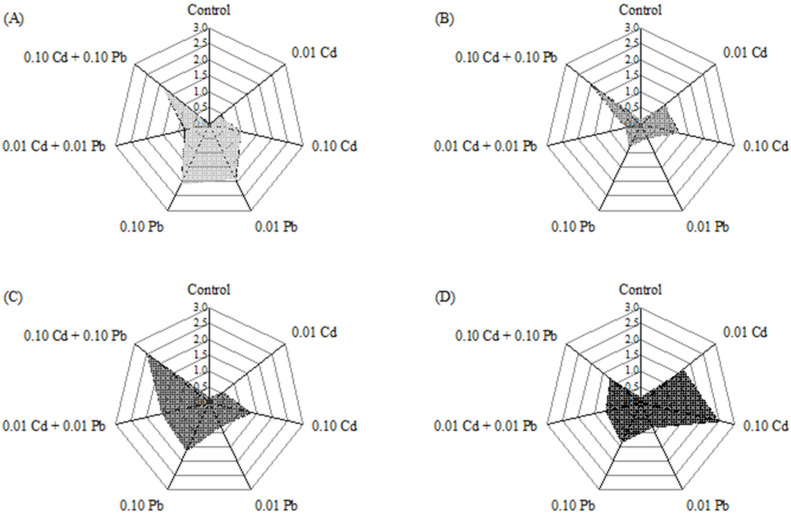
Integrated biomarker response (IBR) of all biomarkers measured in the hepatopancreas of *M. nipponense* for different exposure protocols; (**A**): 7 d, (**B**): 14 d, (**C**): 21 d and (**D**): 28 d.

**Figure 5 ijerph-20-00360-f005:**
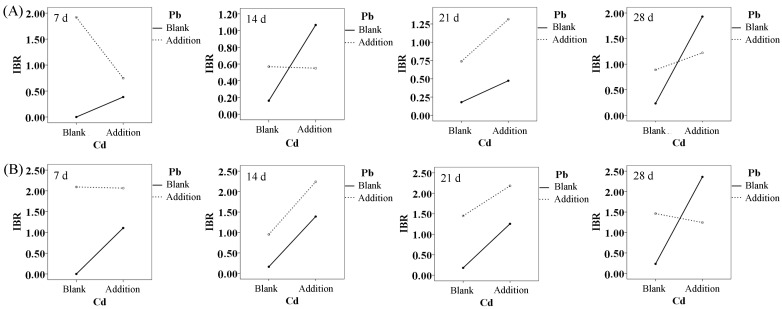
The interaction effects of Cd and Pb in waters at ambient concentrations (**A**): 0.01 mg/L, (**B**): 0.10 mg/L) on hepatopancreas of *M. nipponense* after exposure for 7, 14, 21 and 28 d.

**Table 1 ijerph-20-00360-t001:** Factorial analyses of combined effects of Cd-Pb with different concentrations in waters on IBR in the hepatopancreas of *M. nipponense* during different periods.

Treatment Groups	7 d			14 d			21 d			28 d		
	F-Value	*p*-Value	Partial Eta^2^	F-Value	*p*-Value	Partial Eta^2^	F-Value	*p*-Value	Partial Eta^2^	F-Value	*p*-Value	Partial Eta^2^
Cd (L)	6177.960	<0.001 *	0.999	5895.841	<0.001 *	0.999	5680.666	<0.001 *	0.999	30,891.527	<0.001 *	1.000
Pb (L)	52,084.368	<0.001 *	1.000	89.707	<0.001 *	0.918	14710.502	<0.001 *	0.999	20.047	0.002 *	0.715
Cd + Pb (L)	24,192.692	<0.001 *	1.000	6397.262	<0.001 *	0.999	598.547	<0.001 *	0.987	13,968.046	<0.001 *	0.999
Cd (H)	11,635.937	<0.001 *	0.999	47,427.871	<0.001 *	0.782	24,546.321	<0.001 *	1.000	27,286.311	<0.001 *	1.000
Pb (H)	93,201.984	<0.001 *	1.000	20,159.702	<0.001 *	0.953	36,270.306	<0.001 *	1.000	98.499	<0.001 *	0.925
Cd + Pb (H)	12,866.365	<0.001 *	0.999	25.317	0.001 *	0.871	881.853	<0.001 *	0.991	41256.759	<0.001 *	1.000

Note: * indicated significant difference (*p* < 0.05); Partial Eta^2^ indicated the contribution of each factor to variation. L and H represented the corresponding concentrations of 0.01 mg/L and 0.10 mg/L, respectively.

## Data Availability

Not applicable.

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
