# Peer review of "Oxidative Stress of Cadmium and Lead at Environmentally Relevant Concentrations on Hepatopancreas of Macrobrachium nipponensis and Their Mixture Interactivity: Implications for Water Quality Criteria Amendment"

_ijerph, 2022, doi:10.3390/ijerph20010360_

Round 1

Reviewer 1 Report

In this study, the authors investigate the impact of Cadmium (Cd) and lead (Pb) singly or in combination at two concentration levels and through 28 days, on some selected biomarkers measured in the hepatopancreas of the Asiatic freshwater shrimp Macrobrachium Nipponense. In particular, they measured the activity of some enzymes (Superoxide Dismutase, SOD; Catalase, CAT; Glutathione Peroxidase, GPx; Acetylcholine Esterase, ACH) in addition to concentrations of Metallothionein (MT) and Malondialdehyde (MAD) as a marker for lipid peroxidation. In their evaluation and interpretation of data, the authors apply a so-called “Integrated Biomarker Response” which should reflect the integrated response of all biomarkers together across the applied concentrations and different times of exposure. The result of this integrated approach is then linked to the “ecological health” and the improvement of water quality criteria in the field. While the selection of the tested biomarkers makes sense, the study suffers from some major deficiencies.

1.      First of all, I think that a major flaw of this study is the lacking information about accumulated Cd and Pb concentrations in the midgut gland of exposed shrimps. It is difficult to draw reliable conclusions about metal absorption and the response of biomarkers to these metals without knowing the metal concentrations in the organ in which the biomarkers were measured. The differential and concentration-dependent response patterns of biomarkers at different times of exposure could be the result, for example, of differential absorption and uptake rates of the two metals: an information that could only be achieved by knowing the actual metal concentrations of the two metals in the hepatopancreas. Moreover, I think that more knowledge would also have been useful with regard to the actual metal concentrations in the exposure solutions, instead of relying exclusively on nominal concentrations.

2.      A second point of wariness to me is the application of the so-called “Integrated Biomarker Response” (IBR). While I think that the calculation of such an index could make sense under certain preconditions, I’m a bit skeptical about its validity in the present case, without any knowledge of metal concentrations and their uptake patterns in the hepatopancreas of these shrimps (see my comments above).

3.      Another point of weakness of this study is the fact that the authors missed to explicitly cite the authors who originally introduced the IBR. In particular, the authors should have cited, in this context, two important publications. The first one if from Beliaeff and Burgeot (Beliaeff and Burgeot, in: Environ. Toxicol. Chem. 21 (6), 1316-1322), the second one is from Sanchez et al. (2013), in: Environ. Sci. Pollut. Res. (2013) 20, 721–2725. I appreciate the fact that the authors have apparently contributed much to the application of this index. However, it is a common procedure in the scientific community not to omit citations of authors who have contributed significantly and originally to the application of an important approach.

4.      To add another point, I think that the authors should have explained more carefully why and how Cd and Pb (two metals that do not directly participate in Fenton reactions) may contribute to oxidative stress in an organism. May I suggest, in this context, to consider the following publication: Patra et al., Oxidative Stress in Lead and Cadmium Toxicity and Its Amelioration, in: Veterinary Medicine International Volume 2011, 9 pp.

5.      Last but not least, I think that the manuscript would need a complete revision of style and grammar, perhaps with the help of a native English speaker or of a person who perfectly speaks English.

Reviewer 2 Report

The paper focuses on the oxidative stress effects of the dissolved metal ions on M. Nipponensis exposed in lab for 28 days. Using this typical work, the authors attempt to systematically evaluate the oxidative stress-related damage and their interactivity of different metal ions in crustanceans. In a word, this work is interesting and has a certain signigicant implacations in environment managament field. The experimental data of whole paper is abundant and the analysis methods sound  good. Based on those above, I recommend this draft to publish here. Some minor issues:

1) There are some spelling mistakes, e.g., the irons in the line 110 of page 3 should be ions. Please check the whole paper.

2) Some symbols are missing in Fig. 5? Where is B? The symbol A represents the "addition" or concentration 0. 01 mg/L? Please label it with different symbol.  
